# Design, Fabrication, and Preliminary Validation of Patient-Specific Spine Section Phantoms for Use in Training Spine Surgeons Outside the Operating Room/Theatre

**DOI:** 10.3390/bioengineering10121345

**Published:** 2023-11-23

**Authors:** Marina Carbone, Rosanna Maria Viglialoro, Sara Stagnari, Sara Condino, Marco Gesi, Michelangelo Scaglione, Paolo Domenico Parchi

**Affiliations:** 1Department of Information Engineering, University of Pisa, 56126 Pisa, Italy; sara.condino@endocas.unipi.it; 2EndoCAS Center, Department of Translational Research and New Technologies in Medicine and Surgery, University of Pisa, 56126 Pisa, Italy; rosanna.viglialoro@endocas.unipi.it; 3Department of Orthopaedics and Trauma Surgery, University of Pisa, 56100 Pisa, Italy; sara.stagnari@gmail.com (S.S.); michelangelo.scaglione@unipi.it (M.S.); paolo.parchi@unipi.it (P.D.P.); 4Center for Rehabilitative Medicine “Sport and Anatomy”, University of Pisa, 56121 Pisa, Italy; marco.gesi@med.unipi.it; 5Department of Translational Research and New Technologies in Medicine and Surgery, University of Pisa, 56126 Pisa, Italy

**Keywords:** spine surgery, pedicle screw fixation, patient-specific phantom, 3D printing, surgical training, surgical simulation, learning curve

## Abstract

Pedicle screw fixation (PSF) demands rigorous training to mitigate the risk of severe neurovascular complications arising from screw misplacement. This paper introduces a patient-specific phantom designed for PSF training, extending a portion of the learning process beyond the confines of the surgical room. Six phantoms of the thoracolumbar region were fabricated from radiological datasets, combining 3D printing and casting techniques. The phantoms were employed in three training sessions by a fifth-year resident who performed full training on all six phantoms; he/she placed a total of 57 pedicle screws. Analysis of the learning curve, focusing on time per screw and positioning accuracy, revealed attainment of an asymptotic performance level (around 3 min per screw) after 40 screws. The phantom’s efficacy was evaluated by three experts and six residents, each inserting a minimum of four screws. Initial assessments confirmed face, content, and construct validity, affirming the patient-specific phantoms as a valuable training resource. These proposed phantoms exhibit great promise as an essential tool in surgical training as they exhibited a demonstrable learning effect on the PSF technique. This study lays the foundation for further exploration and underscores the potential impact of these patient-specific phantoms on the future of spinal surgical education.

## 1. Introduction

The anatomy of the spine and its proximity to important neurovascular structures pose a formidable surgical challenge, especially when treating complex deformities or spinal tumors. Research has brought a lot of innovations in surgical approach with the recent development of surgical navigators [1,2] and surgical robots [3,4,5]. Such innovations contribute to a marked improvement in spine-surgery outcomes [3,6,7,8]. Nonetheless, there is a learning curve that must be mastered by the surgeon. Thus, any innovation used at the training stage contributes to improving procedural accuracy and, as such, is welcomed.

Spine-surgeon training takes place during either an orthopedic or a neurological surgery residency program, sometimes followed by an optional spine-surgery fellowship. In Italy and in many other countries, orthopedic spine surgeons receive limited exposure to spine surgery during the residency [9]. On the other hand, neurological surgeons receive an appreciable amount of exposure to spine surgery throughout the residency. At some hospitals in the United States, a resident can choose to focus on spinal conditions during the final year of training but does not have to complete a spine fellowship before performing spine surgery. As underlined by Daniels et al. [10], the two separate training paths (neurosurgery and orthopedic surgery) produce surgeons who receive different spine-surgery exposure, even though there are published guidelines dedicated to spine-surgeon training [11]. Indeed, the rarity of cases and the relatively low incidence of neurosurgical diseases compared with other surgical subspecialties present additional challenges to ensure adequate trainee exposure to surgical pathology. This is why, recently, the merits of a residency program dedicated to spine surgery have been discussed [10]. In the future, spine surgery may benefit from having its own residency training, underscoring the need for innovative training instruments.

The proper placement of transpedicular screws (Pedicle Screw Fixation, PSF) is critical to the outcomes of surgical treatment for several types of spinal problems such as vertebral fractures, degenerative disc pathology, infectious pathology, spondylolisthesis, scoliosis, and spinal tumors. PSF is the gold standard posterior approach for spinal instrumentation techniques to stabilize spine fusions [12]. PSF is a technically demanding procedure that requires intensive training to avoid severe neurovascular complications due to screw misplacement. The incidence of misplacement, as reported in the literature, is very high, being up to 10–40% of cases when free-hand techniques are used [13,14,15,16] and ~15% when guide-based techniques are used [15,16,17].

The learning curve for the placement of pedicle screws (PSs) has been estimated to be 80 screws (or ~25 cases) and the learning curve plateau starts at ~40 screws [18]. The challenge in PSF training is to ensure residents reach this plateau during residency and in the safest environment possible.

Recent advances in bioengineering and in medical imaging have added a training tool to the spinal surgeon’s armamentarium; namely, three-dimensional (3D) printing. High-resolution 3D diagnostic images (CT and MRI) can be processed to enable 3D visualization and the fast production of 3D physical replicas of the elaborated anatomy. In recent years, the surgical training landscape has been enriched by several 3D-printed enabled simulators [19,20]. These 3D-printed simulators can be used for training purposes or for surgical planning purposes [21]. For the latter, high-resolution 4D printing is of utmost importance, while it is not critical for training purposes. Orthopedic simulators are popular in innovative surgical training programs where trainees gain procedural experience in a safe and controlled environment [22,23,24,25,26,27]. Such 3D-printed anatomical models can provide surgeons with the fourth dimension of haptic feedback that can help them to anticipate technical challenges that may be encountered during surgery.

An essential aspect of successful medical training is a physically correct anatomical model that can be used by a trainee [28]. The phantoms should match the morphology, topology, color, texture, and density of the anatomical structure and mimic their behavior so that trainees can efficiently familiarize themselves with the procedural area and acquire the requisite skills [29]. Therefore, it is pivotal to select the task to be simulated and extensively analyze it to define the anatomical elements that should be implemented to obtain a correct simulation of the identified task.

Using these models, surgeons can eventually perform the entire surgical procedure in a stress-free environment and take note of procedural difficulties and necessary safety measures both for training and planning purposes.

In the present study, we designed and fabricated a patient-specific physical simulator with customizable anatomy that could serve to train residents and help them to acquire the skills needed for PSF outside the operating room/theatre. An initial validation with a resident student was reported following his/her achievement of a learning curve. Additionally, the face content and construct validity of the phantoms were tested.

## 2. Materials and Methods

This section contains descriptions of the peculiarities of the developed anthropomorphic phantom, with details on the implementation/fabrication strategy.

Additionally, an initial validation is presented, describing the testing protocol and the strategies to preliminarily validate the phantoms as a training platform.

### 2.1. Phantom Design and Realization

Before starting the design of the anthropomorphic phantom, the primary learning goals of the anthropomorphic phantom design were established. PSF surgery was segmented into tasks and we isolated the tasks that we wanted to target with the developed spine simulator [30,31]. The primary purpose of the phantoms was to use them for the instruction of trainees in the anatomy of the thoracic–lumbar spine and the technique for pedicle screw placement in that spinal section. The following characteristics were deemed to be indispensable:-To simulate the spine instrumentation and the challenges related to this action, our phantoms included patient-specific bone replicas with a correct replication of the cortico–cancellous interface (thoracic and lumbar vertebrae of actual pathological patients); flexible intervertebral discs to mimic intervertebral natural movements; and a flexible anterior longitudinal ligament to hold the vertebrae together, stabilize the spine, and allow physiological motion. An additional feature of our phantoms was the replication of realistic radiodensities for bony structures.-Patient-specific deepness of the operating room/theater. No efforts were made for the accurate replication of muscle structures, as surgical access challenges were not the intended use of the phantoms. In any case, to provide trainees with the opportunity to experience the same challenges as are faced in the confined space of the operating room/theatre, the replica of the spine was sunk in a soft material that replicated the colors and bulkiness of muscles, providing the trainee with a realistic operating field. Moreover, a skin-like covering allowed an accurate simulation of palpation and surgical incisions.

In this study, 4 spine section phantoms were fabricated, with increasing level of complexity. The proposed solution allowed the expert tutors to select a cohort of cases suitable for the training path they wanted to build. In particular, in the present work, three expert surgeons selected four cases, ranging from standard lumbar spondylosis to mild thoracolumbar scoliosis (Figure 1). Those cases were deemed suitable for a novice trainee facing lumbar PSF for the first time.

Specifically, case 1 was a model of a 73-year-old male with spinal stenosis due to lumbar spondylosis without a spine deformity (levels involved: L1–S1), case 2 was a model of a 75-year-old female with grade 2 L4–L5 spondylolisthesis (levels involved: L1–S1), case 3 was a model of a 74-year-old female with mild lumbar scoliosis (levels involved: L1–S1), and case 4 was a model of a 19-year-old female with severe thoracolumbar scoliosis (45° Cobb) (levels involved: T12–S1). Cases 1 and 2 were rated as low-complexity cases, while Cases 3 and 4 were rated as medium-complexity cases. To optimize the training opportunities, all spinal levels were considered for instrumentation during the training session, even if the clinical case primarily pertained to a smaller subset of levels.

The first step in the manufacturing process of the spine simulator was the extraction of the anatomical components from the anonymized CT scan of the spine of the selected patient.

The CT images were processed using the EndoCAS segmentation pipeline developed at our Center on top of the open-source software ITKSNAP 1.5 [32,33]. This pipeline deployed a neighborhood-based region-growing method that allowed for reproducibility of segmentation, with each vertebra being separately segmented. Mesh optimization stages (removal of artifacts, hole filling, simplification, and filtering) were performed to generate the 3D virtual models of the patient-specific bone structures.

Post-processing followed a structured sequence of steps: the removal of isolated pieces, artifacts, and non-manifold features using MeshLab (www.meshlab.com) (accessed on 17 October 2023). (in that order), followed by simplification and smoothing using Blender (www.blender.com) (accessed on 17 October 2023). These final steps served an esthetic purpose, smoothing the anatomy, and were considered to be important because the phantoms were intended for training rather than surgical planning.

The virtual models were then printed using acrylonitrile butadiene styrene (ABS) and a fusion deposition modeling (FDM) 3D printer (Dimension Elite, Stratasys LTD, Rehovot, Israel). In our study, a medium accuracy (0.18 mm of the Z resolution) was deemed sufficient, considering the dimensions of the vertebral bodies and pedicles and the fact that the printed models were to be used for training and not for surgical planning. ABS is a thermoplastic material commonly used in orthopedic simulations [22] because its mechanical, optical, and radiological properties are comparable with those of bone structures [28,34,35,36].

The selection of material and the 3D printing method used took into consideration the structural characteristics of bones. Specifically, emphasis was placed on replicating the mechanical interface between cortical and cancellous bones; this is very important in surgical procedures because it provides tactile feedback to the surgeon, thereby aiding the assessment of procedural correctness.

The chosen printing approach facilitated the faithful reproduction of the density proportions observed between cortical and cancellous bones. This was accomplished by leveraging the capabilities of a 3D printer equipped with variable infill density control, facilitated through the employment of a slicing software package (GrabCAD Print version 1.72, www.grabcad.com) (accessed on 17 October 2023). Notably, the decision was made to employ the lowest achievable density setting, which corresponded with an infill of 16%. Additionally, the external wall thickness was kept within the range of 0.8 to 1.3 mm.

As a result of these parameter selections, the inner structure of each vertebral body was fabricated with a beehive-like lattice configuration. Although alternative anatomical infill patterns have been proposed in the existing literature, their suitability is contingent upon a specific application such as the creation of radiological phantoms [37]. Given that the primary purpose of the present study was an accurate emulation of the mechanical interface between cortical and cancellous bones, the geometrical beehive infill pattern was deemed to be sufficient [22].

A synthetic material based on flexible silicone foam (Soma Foama™ Smooth-On Inc., Easton, PA, USA) was chosen to replicate the intervertebral discs because it closely mimics the haptic feedback of biological tissue and is radiolucent for fluoroscopic imaging. In more detail, intervertebral discs were obtained by cutting out a sheet of silicone foam, positioning the discs between adjacent surfaces of the printed bony vertebrae, and customizing them to replicate the thickness of the intervertebral disc. A printed and assembled model of the spine section is shown in Figure 2. Finally, rubber bands and another silicone material (Dragon Skin™ FX-Pro™ Smooth-On Inc., Macungie, PA, USA) were used to mimic the anterior longitudinal ligament. This synthetic anterior longitudinal ligament can stabilize the spine and allow for physiologic motion. The final phantom included a replica of the entire thoracolumbar spine embedded in soft polyurethane foam (FlexFoam-iT™ Smooth-On Inc., Macungie, PA, USA). The latter was used to replicate paravertebral soft tissues that can impede visualization during screw placement, allowing the use of retractors during the simulated procedure, as well as the need for skeletonization of vertebral pedicles. For a high-fidelity simulation, a skin-like covering was placed on the final phantom [19,22,38]. To mimic skin tissue properties, a silicone mixture based on Ecoflex 0010 silicone rubber (Smooth-On Inc.) and additives were used to manufacture the skin-like covering in thin sheets. The final phantom is shown in Figure 3.

### 2.2. Testing Setup and Structure

The simulator underwent quantitative and qualitative validation tests.

Quantitative evaluation: One fifth-year resident (named R0) from the Cisanello University Hospital in Pisa was enrolled in this study. R0 had a limited amount of prior experience in placing PSs; specifically, R0 had never placed a spinal screw on a live patient and had only assisted with greater than 10 but fewer than 20 spinal surgeries.

The experimental setup consisted of one of the four anthropomorphic phantoms (chosen according to the difficulty required for the testing session), which was placed on a fixed-height surgical table, along with the demonstration version of the orthopedic surgical equipment (provided by Medtronic Medical Device Company, Dublin, Ireland, (www.medtronic.com)) (accessed on 17 October 2023) required for the selected cases. The equipment included a set of screws ranging from 4.5 to 6.5 mm in diameter and from 40 to 50 mm in length. Before each test session, an expert spine surgeon presented the surgical case, which was simulated by the corresponding phantom. This presentation included a description of the patient’s radiological images and an introduction to the free-hand technique for PSF, along with discussions of the risks, complications, and post-procedure interpretations. An expert orthopedic surgeon supervised all trials with varying levels of involvement, depending on the testing session, as described below.

The study protocol involved the following steps:Identification of the longitudinal surgical access point, determined after locating the spinous processes through deep palpation (Figure 4).Exposure of anatomical landmarks such as facet joints, transverse processes, and the lateral portion of the pars interarticularis following incision, divarication, and the removal of soft tissue from the exposed surface (Figure 4).Creation of the cortical breach and placement of the bone probe after the identification of the entry point. The latter corresponded with the point where the major axis of the transverse process met the line passing through the lateral margin of the superior facial joint (Figure 5).Navigation of the bone probe into the pedicle along the ideal trajectory. This was achieved by aiming for the contralateral transverse process, thus aligning the screw with the superior endplate. To assess the integrity of the hole walls, a probe was inserted (Figure 5).Insertion of the selected screw (ensuring it passed through the canal in the pedicle and affected 2/3 of the vertebral body’s depth (Figure 5).Verification of the PS placement under RX control (Figure 6).Classification of the PS placement, according to the degree of possible pedicle wall violation under CT control (Figure 6).

R0 was tasked with performing PS placements on six phantoms of increasing complexity across three different training sessions (two phantoms per session). The evaluation of R0’s performance was based on recorded PS placement times and post-simulation CT assessments, allowing for the generation of a learning curve.

Further details are as follows:-In the first session, R0 worked on Phantoms 1 and 2 (simulating low-complexity level cases) but under active supervision (the expert surgeon tutored him/her for each screw insertion, guiding R0 and putting hands on the phantom).-In the second session, R0 repeated the work on Phantoms 1 and 2 (low-complexity level) but under passive supervision (the expert surgeon guided R0 by talking to him/her and eventually advising and/or correcting R0’s gestures, but without putting hands on the phantom).-In the third session, R0 worked on Phantoms 3 and 4 (simulating medium-complexity cases) without any supervision; that is, the expert surgeon was present but did not intervene during the session. Two performance measurements were conducted. These were PS placement accuracy, evaluated by the degree of pedicle wall violation on CT images according to the classification of Gertzbein and Robbins [39], and the time taken to place each implanted screw.

The Gertzbein and Robbins classification is graded from A to E. A represents perfect interpeduncular localization, B represents a peduncular breach of <2 mm, C represents a peduncular breach of 2 to <4 mm, D represents a peduncular breach of 4 to <6 mm, and E represents a peduncular breach of >6 mm. In the present study, the last two categories were combined into category D for pedicle wall violations > 4 mm (that is, a peduncular breach of >4 mm). According to Gertzbein and Robbins, a PS placement grading of A or B is considered to be accurate [39].

Qualitative evaluation: For a phantom to be used to assess competence, its validity and reliability as an effective training platform must be determined.

Here, the validity of a phantom refers to whether or not it is used effectively to impart the intended knowledge or to evaluate the desired skill [40,41,42]. Face validity pertains to how realistic the phantom is; that is, does it accurately represent what it is intended to emulate? Content validity addresses phantom reliability and involves experts in the subject matter of the training device and evaluates the appropriateness of the phantom as a teaching tool. It addresses the question: does use of the phantom allow the intended knowledge to be conveyed realistically?

Construct validity is one of the most crucial aspects to assess. It determines whether the phantom is a valid tool to evaluate whether experienced surgeons outperform inexperienced ones. This can be quantitatively assessed.

In the last part of the study, we enrolled three expert spine surgeons and six novices (one medical student and five orthopedic residents (one first-year resident, two fourth-year residents, and two fifth-year residents)) to test the phantom for face, content, and construct validity [40]. Subjective feedback regarding the phantom’s realism and effectiveness in facilitating the intended simulation was gathered from all nine participants during an open brainstorming debriefing at the end of the session.

During the brainstorming debriefing, four statements were presented, each leading to lively discussions. Detailed notes were recorded.

The statements were as follows:The anthropomorphic phantom accurately replicated the surgical field and necessary anatomy for the simulation of posterior pedicle screw insertions.The anthropomorphic phantom provided a surgical field closely resembling an actual one in terms of confined space, anatomical structure footprint, and visibility.The feedback on hands and surgical instruments during the surgical tasks was realistic.The anthropomorphic phantom, when arranged in a simulation course with increasing complexity, was a valuable platform for the teaching of how to perform posterior pedicle screw fixation.

Each of the nine participants also tested the anthropomorphic phantom by inserting at least 4 screws (performing one complete level of instrumentation). The positioning of the screws was then blindly evaluated to determine if the expert surgeons performed better than the novices.

## 3. Results

### 3.1. Quantitative Evaluation

R0 placed a total of 57 pedicle screws across three training sessions.

This number aligned with the defined interval of Gonzalvo et al. [18], which indicated the number of screws needed to reach a learning curve plateau. This observation was further confirmed by the plot of R0’s results (Figure 7).

Following Gertzbein and Robbins’ classification, 84.2% (48/57) of the screws received an A grade, while 10.5% (6/57) were rated as B, 3.5% (2/57) as C, and 1.7% (1/57) as D. A detailed breakdown of each session’s results is presented in Table 1, which provides a comprehensive view of R0’s performance improvement over time. Figure 7 illustrates a noteworthy enhancement of the success rate of pedicle screw placement with accumulating experience. Notably, there was a marked increase in overall PS success between Trials 1 and 3. The most noticeable improvement was seen between Trials 1 and 2, after which the curve reached a plateau by Trial 3.

R0 showed a decrease in PS placement time with each implanted screw, with the median time rapidly decreasing from 12 min per screw during the first phantom to 6.87 min per screw for the second phantom (during Session 1), or 7 min per screw (first phantom) to 3.27 min per screw in Session 2 and 3.07 min per screw in Session 3. Together with the learning curve (Figure 8I), we executed a CUSUM analysis for the time per screw data to determine the point where the resident reached a plateau [43]. The CUSUM (Equation (1)) was executed using MATLAB R2023a (www.mathworks.com) (accessed on 17 October 2023).
(1)$$Cusum_{s}\sum_{j=1}^{i}(X_{j}-\bar{X}),$$
where the mean $\bar{X}$ is the mean time of the first session (first 22 screws); that is, the area where data were more dispersed [43].

The CUSUM score analysis evidenced that the number of PS placements required by R0 to achieve mastery was N_mast_ = 40 (Figure 8II). The mean time per screw reached in the last session was more comparable with that achieved by expert spine surgeons on patients [44,45].

### 3.2. Qualitative Evaluation

Face and Content Validity: The consensus among the nine participants was that the spine phantoms closely replicated the surgical field, particularly in terms of the soft tissue’s bulkiness and the challenges associated with identifying vertebral anatomy. The 3D-printed vertebrae allowed for successful pedicle screw instrumentation with correct haptic feedback, providing an accurate representation of the cortical/cancellous bone interface. All the participants—in particular, the expert surgeons—stressed the fact that a training path should include sessions on these kinds of phantoms to bring part of the learning curve outside the operating theatre/room. Additionally, all the participants were of the opinion that the phantom demonstrated a high-fidelity representation of bony structures under fluoroscopy.

Construct Validity: The use of the phantoms allowed a differentiation of the experience level of the participants. Thus, expert surgeons consistently outperformed their less-experienced counterparts, achieving a perfect score of 100% (12/12) with grade A screws. In contrast, among the less-experienced trainees, 34.2% (14/41) received a grade of A, 39% (16/41) were graded as B, 2.4% (1/41) received a grade of C, and 24.4% (10/41) received a grade of D.

## 4. Discussion

Pedicle screw fixation (PSF) necessitates a dedicated training program due to the imperative need for mastery of the technique and a deep understanding of anatomy. This is particularly critical for trainee spine surgeons who may not have this surgical skill, especially given the potential life-threatening consequences of errors that can occur during pedicle screw instrumentation [13,14,15].

Although cadaveric training offers a safe low-stress environment for surgical practice, obtaining an adequate number of cadavers to teach PSF is challenging [9]. Moreover, cadavers seldom present pathologies or deformities tailored to the expertise level of the trainee.

In general, training through simulation has been demonstrated to be a good instrument for challenging surgeries [44].

Our study sought to demonstrate the effectiveness of 3D phantoms of spine segments to enhance the accuracy and efficiency required for PSF. The use of our anthropomorphic phantoms also revealed a learning curve effect, resulting in fewer pedicle wall violations in the final training session. The median percentage of pedicle violations markedly decreased with the training time with the phantoms, dropping from 28% to 3%. Additionally, the time required to complete the instrumentation decreased over the course of the training. Drawing from the experience of R0, the median time needed to place a screw decreased from an initial ~12 min to ~3 min in the final session, reaching a plateau at N_mast_ = 40 screws.

Some studies in the literature present patient-specific spinal phantoms [20,27,44,46]; to the best of authors’ knowledge, none have evaluated the effectiveness of surgical training on such tailored simulators together with a validation of the simulator. The proposed simulators often limit themselves to directly printing the vertebral bodies, sometimes without even separating each vertebra, and insufficient attention is given to the necessity of enriching the simulation with soft tissues. Clifton et al. [20] introduced an intriguing simulator and demonstrated its feasibility and economic viability by adding a foam layer covering to simulate surgical access.

Park et al. [47] proposed a study to evaluate the educational effects of training on a 3D-printed life-sized spinal model for inexperienced surgeons, reporting the existence of a learning effect during the repetition of the procedure on various models with an increase in accuracy of screw placement and decrease in pedicle infractions. However, in their study, the phantom lacked the realism necessary for a comprehensive evaluation of its usefulness in terms of surgical field reproduction. Hong et al. [27] presented a well-structured trial on a 3D-printed spine; the simulated soft tissues did not entirely envelop the bony tissue, leading to a lack of realism in accurately mimicking the surgical environment. Despite this limitation, the researchers were able to observe a discernible learning effect among four residents, each tasked with placing a total of 18 screws over three sessions. Their analysis focused on the PSF accuracy and procedure time according to repeated training. However, the training was insufficient to evaluate a complete learning curve.

The training phantoms fabricated in the present study introduced a fourth dimension into the residency training program, complementing the standard training approach. During the phantom laboratory, residents could familiarize themselves with the surgical techniques as well as the characteristics of various steps in spinal surgery such as skeletonization, the identification and exposure of pedicle screw entry points, and the orientation of pedicle screws according to the correct trajectory. All this took place in a non-stressful and risk-free atmosphere. Moreover, experienced surgeons can also benefit from phantom simulation. Complex clinical cases such as congenital deformities, reinterventions in cases of failed previous stabilization procedures, and severe scoliosis with marked rotation and vertebral deformities can be faithfully replicated using our phantoms. Surgeons dealing with scoliosis are well-acquainted with the intraoperative challenges inherent in the surgical treatment of these complex deformities, including vertebral rotation, absent or dysmorphic pedicles, and segmentation abnormalities, which are all typical features of scoliotic spines that alter anatomical landmarks for transpedicular screw insertion. Dedicated phantoms allow surgeons to preoperatively familiarize themselves with the technical complexity associated with challenging procedures, thereby enhancing their skills. 

Regarding the limitations of the study, we recognize that the sample size was too small to allow for a meaningful statistical analysis and the qualitative assessment lacked a structured questionnaire and Likert-scale evaluation. Nonetheless, even though R0 may not have had prior experience of placing screws and may have only assisted with a limited number of live spinal surgeries, he/she accumulated significant experience in terms of cognitive and decision-making skills throughout their residency program. This led to a more rapid acquisition of the required skills during the spinal training.

Iterations of this study will involve a larger participant cohort to provide a more comprehensive assessment of the learning curve, including a follow-up evaluation of trained participants and a wider case library to enhance the trainer surgeons’ possibility of building a training path tailored to the novice’s starting experience and learning objectives.

Despite these limitations, our study indicated that our phantoms have promise to be an effective tool to train surgeons in vertebral instrumentation techniques with pedicle screws. These findings underscore the importance of establishing supplementary training programs for the education of trainee spine surgeons. However, we acknowledge the need for further studies involving a larger cohort of participants (residents and expert surgeons) and a follow-up assessment of real patients after training to conclusively confirm the learning effect suggested by this initial evaluation study. 

We are aware of the fact that the cost of our phantom is high, being comparable with or slightly lower than that for acquiring a cadaver or a cadaveric sample. The meticulous selection of materials for all the phantom components (from the printing material to the skin-like silicone texture) has led to higher consumable costs and the fabrication process still requires a significant amount of time from skilled personnel. Thus, the cost was around EUR 300 of materials for a phantom, including up to 10 vertebral levels; this cost included a mean of 24 h of printing (the machining time also had a cost). An additional cost of approximately 4 man hours should be considered separately.

Nevertheless, when compared with cadaver costs (ranging from EUR 2000 to 3000 per sample), we contend that our phantom offered the advantage of tailoring the spinal anatomy and customizing the training experience. Furthermore, with our phantom, all surgical instrumentation (including screws) were retrieved at the end of each session, eliminating the expenses associated with sterilizing equipment and environments. An additional noteworthy point is that a phantom laboratory can be set up anywhere without insurance concerns for participants. Indeed, an industrialization of the procedure could significantly lower production costs. In fact, a university spin-off (e-Spres3D s.r.l.) is currently working to industrialize the process and commercialize the proposed solution. 

## 5. Conclusions

This paper focused on the conceptualization and fabrication of a 3D-printed spinal phantom for training in posterior screw fixation procedures, and reported the initial evaluation of its effectiveness as a training tool. The aim was to develop a structured training program based on patient-specific phantoms to help novice surgeons to reach a plateau in their learning curve in a controlled and safe environment without the risk of an actual surgical setting.

The analysis of the learning curve obtained from the trainee involved in the study showed promising results, with an evident plateau both in accuracy and time per screw after 40 screws. These results should be confirmed with a larger study.

## Figures and Tables

**Figure 1 bioengineering-10-01345-f001:**
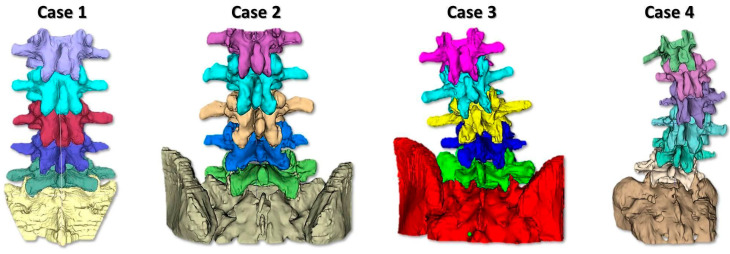
The segmented vertebral anatomy of the four simulated surgical cases.

**Figure 2 bioengineering-10-01345-f002:**
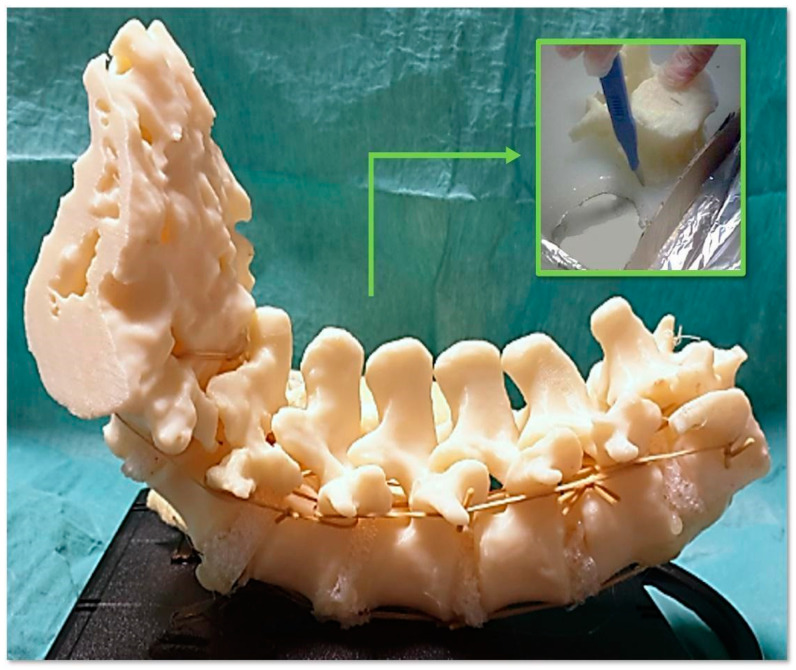
Case 4: printed model of the spine section already assembled with the intervertebral disks.

**Figure 3 bioengineering-10-01345-f003:**
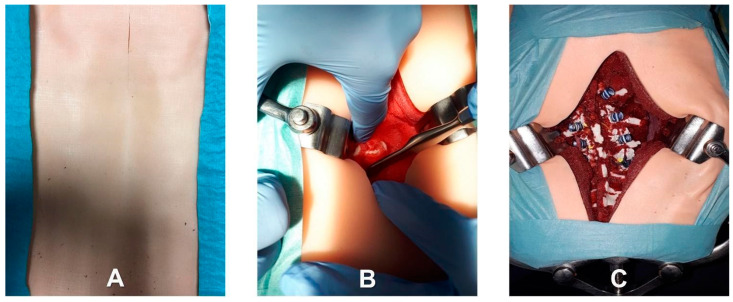
Photographs of the final spine section phantom as seen from the surgeon’s point of view: (**A**) exterior aspect; (**B**) phantom during the execution of the surgical task; (**C**) phantom completely instrumented.

**Figure 4 bioengineering-10-01345-f004:**
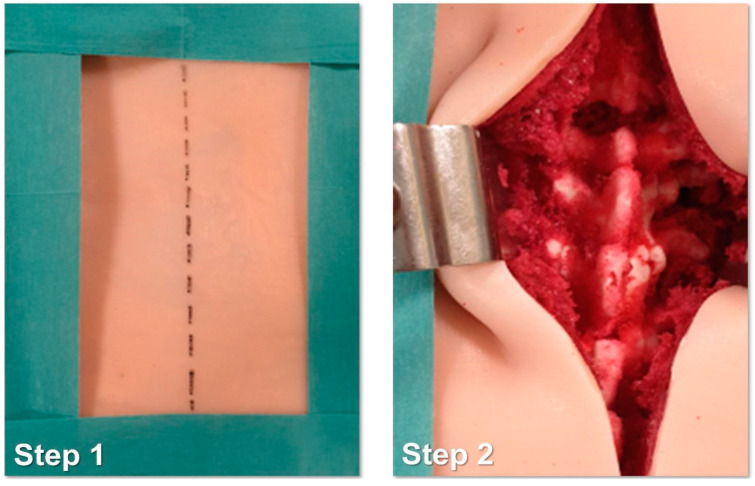
Step 1 and 2 of the simulated surgical procedure.

**Figure 5 bioengineering-10-01345-f005:**
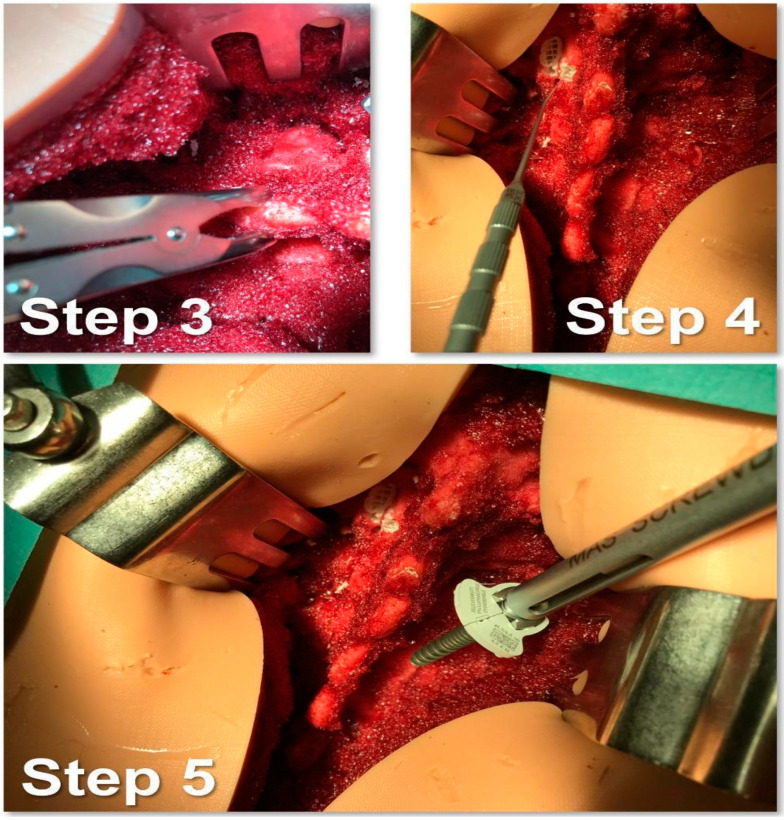
Step 3, 4, and 5 of the simulated surgical procedure.

**Figure 6 bioengineering-10-01345-f006:**
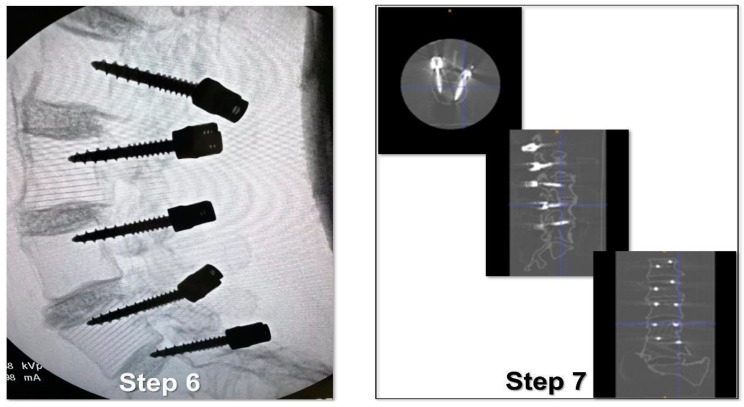
Step 6 and 7 of the simulated surgical procedure.

**Figure 7 bioengineering-10-01345-f007:**
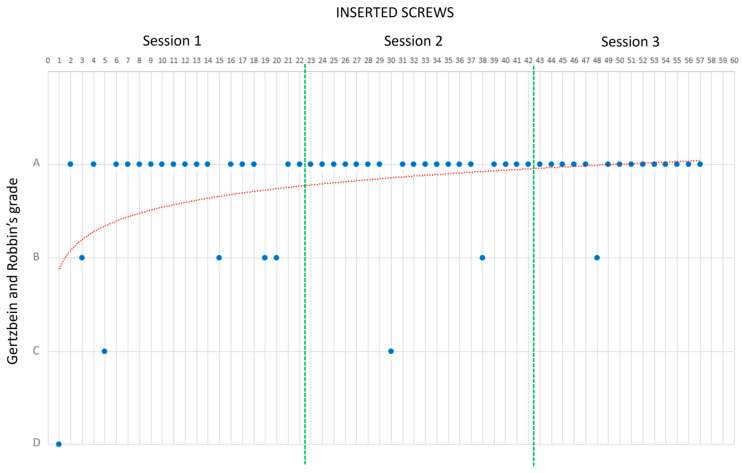
R0’s learning curve, plotted for screw positioning grading. Blue dots are the screws grades; red, dotted, line is the interpolated learning curve; the green bar separates the three ssessions results for better understanding.

**Figure 8 bioengineering-10-01345-f008:**
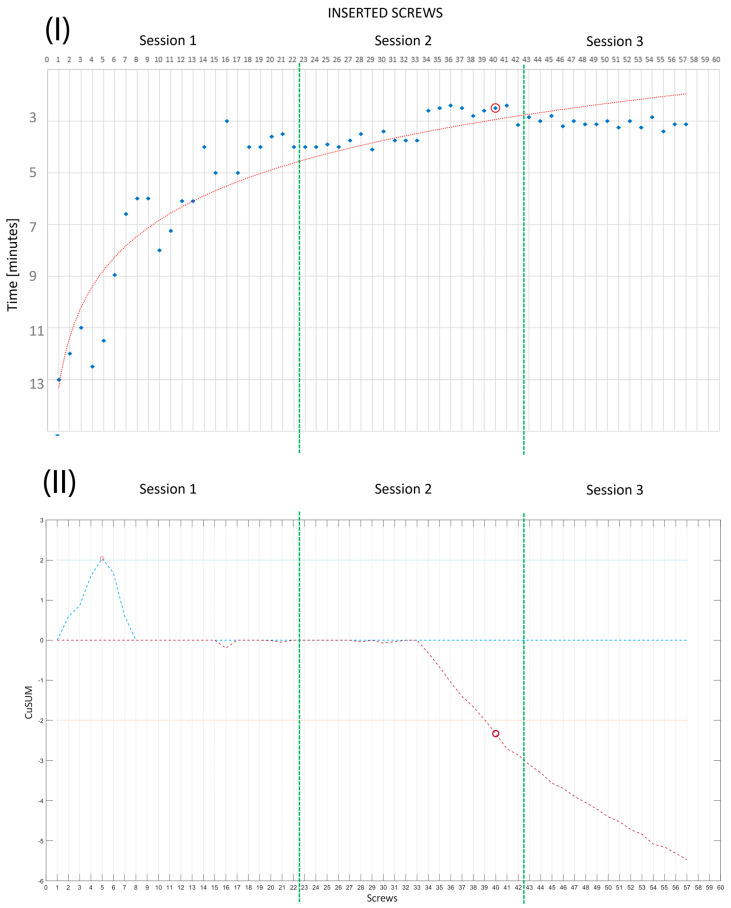
The green bar separates the three sessions results for better understanding. (**I**) R0’s learning curve plotted for time per screw (minutes);blue dots are the time per screw; red, dotted, line is the interpolated learning curve; and (**II**) the relative CUSUM analysis (positive—blue and negative—red) evidencing that the plateau was reached at screw #40 (red circle).

**Table 1 bioengineering-10-01345-t001:** Percentage of screws per grade and per training session, representative of R0’s performance according to Gertzbein and Robbins’ classification.

	Percentage of Screws per Grade			
Session	Grade A	Grade B	Grade C	Grade D
1	73%	18%	5%	5%
2	90%	5%	5%	
3	93%	5%		

## Data Availability

Data are available upon reasonable request. Please contact the corresponding author.

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
