# Peer review of "Design, Fabrication, and Preliminary Validation of Patient-Specific Spine Section Phantoms for Use in Training Spine Surgeons Outside the Operating Room/Theatre"

_bioengineering, 2023, doi:10.3390/bioengineering10121345_

Round 1

Reviewer 1 Report

Comments and Suggestions for Authors

The authors constructed a patient-specific model designed for pedicle screw fixation (PSF) training. And in combination with 3D printing technology and imaging radiology, multiple spine models were constructed to enable learning for PSF training outside the surgical room. Overall, this PSF surgical training model construction technique has certain applications and research value, but still needs major revisions before it can be considered for publication.

1. The research on building precision 3D printed products based on imaging methods is also mentioned in the article "Emerging 3D Bioprinting Applications in Plastic Surgery. Biomaterials research, 27(1), 1". It is recommended that the authors make additions, regarding the significance of high precision 3D printed products in Introduction.

2. To facilitate reader comprehension, it is recommended that the authors improve the quality of the manuscript's writing.

3. As described by the authors, four cases of spinal disorders were constructed in the present study. It should be necessary for the authors to add an explanation of the reasons and significance of selecting these four spinal deformity models. Can these only four cases meet the PSF surgical needs of surgical learners coping with different clinical patients?

4. Is it possible to add new specific case models of spinal deformity in the follow-up studies that the authors expect? It would probably make the study more scientifically valid.

5. The authors' Discussion section is too long. The section should highlight the content, highlights, shortcomings, and potential value of the authors' research. Appropriate streamlining of the Discussion section would have contributed to the readability of this manuscript.

Comments on the Quality of English Language

 Minor editing of English language required

Reviewer 2 Report

Comments and Suggestions for Authors

Ref. No.: bioengineering-2694842

Subject: Decision on Manuscript: Development of Patient-Specific Spine Phantoms and Initial Validation to Bring Part of the Learning Curve Outside the Surgical Room

Journal: Bioengineering

Dear Editor,

I would like to thank you for the invitation to collaborate to review process of article “Development of Patient-Specific Spine Phantoms and Initial Validation to Bring Part of the Learning Curve Outside the Surgical Room”. I recommend that is necessary a major revision of manuscript. Some comments are described below:

English should be improved in all manuscript.

Abstract should be clarifying the novelty of this research. In addition, more quantitative information should be added.

As underlined by [10] the two separate training path (neurosurgery and orthopedic surgery) produces surgeons that receive different spine surgery exposure, even if there are published guidelines dedicated to spine surgeon training [11].” The name of first author added to et al. should be included before [10].

In the introduction, the authors should be added more information based on data from literature about the importance and innovations on 3D simulations using in orthopedic area and Orthopedic simulators.

“Realistic anatomical reproduction of the physical components. An essential aspect of successful medical training is a physically correct anatomical model that can be used by a trainee [29-31]. The physical anatomical models should match the morphology, topology, color, texture, and density and mimic the behavior of the anatomical structure so that the trainees can familiarize themselves with the procedural area and 110 gather skills efficiently [32]. Therefore, it is pivotal to select the task to be simulated and deeply analyze it to define the anatomical elements that should be implemented to obtain a correct simulation of the identified task. Therefore, as our aim was to simulate the spine instrumentation and the challenges related to this action, our simulator includes patient-specific bone replicas with a correct replication of the cortico-cancellous interface (thoracic and lumbar vertebrae of actual pathological patients); flexible intervertebral discs to mimic inter-vertebral natural movements; and a flexible anterior longitudinal ligament to hold the vertebrae together, stabilize the spine and allow physiologic motion. A further feature of our patient-specific physical simulator is the replication of realistic radiodensities for bony structures.” These parts should be replaced to introduction to emphases the importance of uses these techniques.

Terms like in vivo should be in italic font.

Table 1 should be re-written as average and deviation. Statistical analysis is extremely important to perform.

“Few studies in the literature present patient-specific spinal phantoms [22, 45, 46] and none have attempted to evaluate the effectiveness of surgical training on such tailored simulators.” This phrase should be re-written to “and as the authors knowledge’s none have attempted to evaluate the effectiveness of surgical training on such tailored simulators.”

In all discussions and manuscript, the plural first person should be avoided.

“The costs associated with the proposed phantom remain relatively high, and comparable or slightly low to the expenses of a cadaveric sample. The meticulous selection of materials for all components (from the printing material to the skin-like silicone texture) has led to higher consumable costs, and the fabrication process still requires a significant amount of time from skilled personnel so the cost is around 300€ of materials for a phantom including up to 10 vertebral levels, this cost includes a mean of 24 hours of printing (machining time has a cost) while an additional cost of approximately 4 man-hours should be considered separately” The authors should be compared the differences of cost (amount) of these two procedures.

Conclusions should be added.

Comments on the Quality of English Language

English should be improved in all manuscript.

Reviewer 3 Report

Comments and Suggestions for Authors

PLEASE SEE MY REVIEW (CONTAINING 2 WORD DOCUMENTS) IN THE UPLOADED ZIP FILE.

Comments on the Quality of English Language

PLEASE SEE MY REVIEW (CONTAINING 2 WORD DOCUMENTS) IN THE UPLOADED ZIP FILE.

Round 2

Reviewer 1 Report

Comments and Suggestions for Authors

accept

Reviewer 2 Report

Comments and Suggestions for Authors

Ref. No.: bioengineering-2694842-v2

Subject: Decision on Manuscript: Development of Patient-Specific Spine Phantoms and Initial Validation to Bring Part of the Learning Curve Outside the Surgical Room

Journal: Bioengineering

Dear Editor,

I would like to thank you for the invitation to collaborate to review process of article “Development of Patient-Specific Spine Phantoms and Initial Validation to Bring Part of the Learning Curve Outside the Surgical Room”. My recommendation is described below:

The authors did the required all corrections and the manuscript is publishable in current version.

Comments on the Quality of English Language

English is ok.